# Different cellular and molecular responses of Bovine milk phagocytes to persistent and transient strains of *Streptococcus uberis* causing mastitis

Anyaphat Srithanasuwan[1,2], Ynte H. Schukken[2,3,4], Noppason Pangprasit[1,5], Phongsakorn Chuammitri[1,6], Witaya Suriyasathaporn[1,6,7]*

1 Faculty of Veterinary Medicine, Chiang Mai University, Chiang Mai, Thailand, 2 Department of Animal Sciences, Wageningen University, Wageningen, the Netherlands, 3 GD Animal Health, Deventer, the Netherlands, 4 Department of Population Health Sciences, Utrecht University, Utrecht, the Netherlands, 5 Akkhraratchakumari Veterinary College, Walailak University, Nakhon Si Thammarat, Thailand, 6 Research Center of Producing and Development of Products and Innovations for Animal Health and Production, Chiang Mai University, Chiang Mai, Thailand, 7 Asian Satellite Campuses Institute-Cambodian Campus, Nagoya University, Nagoya, Japan

* suriyasathaporn.witaya.y3@f.mail.nagoya-u.ac.jp, suriyasathaporn@hotmail.com

## Abstract

*Streptococcus uberis* is frequently isolated from milk collected from dairy cows with mastitis. According to the host's immunity, bacterial virulence, and their interaction, infection with some strains can induce persistent subclinical inflammation, while infection with others induces severe inflammation and transient mastitis. This study compared the inflammatory response of milk-isolated white blood cells (mWBCs) to persistent and transient *S. uberis* strains. Quarter milk samples were collected aseptically for bacterial culture from all lactating cows once a week over a 10-week period. A transient and noncapsular strain with a 1-week intramammary infection duration was selected from this herd, while a persistent and capsular *S. uberis* strain with an intramammary infection longer than 2 months from our previous study was selected based on an identical pulse field gel electrophoresis pattern during the IMI episode. Cellular and molecular responses of mWBCs were tested, and the data were analyzed using repeated analysis of variance. The results showed a higher response in migration, reactive oxygen species generation, and bacterial killing when cells were stimulated with transient *S. uberis*. In contrast, the persistent strain led to increased neutrophil extracellular trap release. This study also highlighted several important molecular aspects of mWBCs. Gene expression analyses by real-time RT-PCR revealed a significant elevation in the expression of Toll-like receptors (TLR-1, TLR-2, TLR-6) and proinflammatory cytokines (tumor necrosis factor-alpha or TNF-α) with the transient strain. Additionally, *Streptococcus uberis* capsule formation might contribute to the capability of these strains to induce different immune responses. Altogether, these results focus on the immune function of activated mWBCs which demonstrate that a transient strain can elicit a stronger local immune response and, subsequently, lead to rapid recovery from mastitis.

**Data Availability Statement:** All relevant data are within the paper and its Supporting Information files.

**Funding:** Thailand Research Fund and Thailand Science Research and Innovation Grant number: PHD/0220/2561 https://rgj.trf.or.th/main/home/ Anyaphat Srithanasuwan.

**Competing interests:** The authors have declared that no competing interests exist.

## Introduction

The duration and severity of mastitis in dairy cows are determined by the combination of the following 3 components: udder defense efficiency, quantity and virulence of invading microbes, and environmental risk factors [1, 2]. An effective immune response can completely remove the invading bacteria and cause a short-term infection, here defined as a transient infection [3]. In the case of the unsuccessful elimination of bacteria, a persistent infection may occur, and a duration longer than 2 months is termed a persistent infection [4, 5].

*Streptococcus uberis* is a major environmental mastitis pathogen, indicating that the source of the infection lies in the cow's environment. However, many molecular epidemiology studies on *S. uberis* intramammary infections in populations of animals indicate contagious transmission [5, 6]. This bacterium has been reported to cause intramammary infection (IMIs) with a wide range of durations [3, 7, 8]. Environmental *S. uberis* strains commonly cause transient IMI, but strains showing contagious behavior can cause either transient IMI, persistent IMI, or both [5]. Consequently, within-strain variation in the immune response to and pathogenicity of *S. uberis* IMI has been reported [9, 10]. As reported in our previous study [11] and in the study of Fu, Zhou, Qiu, Chen, Zhang and Miao [12], IMI caused by *S. uberis* was linked to the expression of different virulence genes, especially *hasA/B* and *lbp*. The gene relating to capsule formation, *hasA/B*, was considered to reduce virulence with a consequent reduction in resistance to the bactericidal action of immune cells. Additionally, our previous study [11] found that the majority of patterns of the for transient *S. uberis* isolated (63.6%) did not include *has*A/B. Given the substantial variation in *S. uberis* strains, the interaction between *S. uberis* strains, IMI, and host immunity is intricate and remains largely uncharacterized.

The innate immune system plays a critical role in response to an IMI and may respond to invading microbes through a combination of natural defensive barriers such as phagocytes, complements, cytokines, and antimicrobial peptides [1, 13]. After recognition of invading microbes, Toll-like receptors (TLRs) trigger signaling pathways that result in the secretion of cytokines and chemokines, including interleukin-1 (IL-1β), interleukin-6 (IL-6), and interleukin-8 (IL-8). These cytokines are involved in recruiting phagocytes to the site of infection and activating pathogen-killing mechanisms [14, 15]. Intramammary phagocytes have varying phenotypes, response profiles, and fates according to how and where they are recruited into tissues from the blood [16]. Previous studies have reported a reduction in the viability and efficacy of activated phagocytes after diapedesis into the mammary gland [17, 18]. These pre-stimulated milk phagocytes have been activated and progressively become functionally exhausted cells due to the interference of milk components with cellular activities, subsequently reducing the antimicrobial activities of these affected cells.

The causal pathogens entering the mammary gland also affect the efficacy of the immune response [19, 20]. Several studies have examined the species-specific immune response against major mastitis pathogens such as *Escherichia coli*, *Staphylococcus aureus*, and *Streptococcus agalactiae* [21–23]. *Escherichia coli*, typically transient in nature, will quickly elicit strong inflammation of the udder and fully activate immune defense [24]. Whereas gram-positive bacteria (such as *S. aureus* and *S. agalactiae*) will slowly elicit a much weaker inflammation and immune response, frequently resulting in chronic infections. Furthermore, between-strain variability within bacterial species has been recently demonstrated, such as acute and persistent *E. coli* [25, 26] and *S. aureus* strains [27, 28]. However, knowledge of the strain-specific immune response of activated milk phagocytes to *S. uberis* is very limited. Hence, the current study was carried out to study the effector functions of milk phagocytes in response to different strains of *S. uberis* that cause mastitis. This study elucidates the strain-specific immune response of milk phagocytes stimulated with a transient or persistent *S. uberis* strain. We

determined gene expression related to the immune response in exposed phagocytes through comprehensive gene expression analysis.

## Materials and methods

### Milk sample collection and bacterial selection

This study was performed from November to December 2019 and authorized by the Chiang Mai University—Animal Care and Use Committee (Ref no. S26/2562). A smallholder dairy farm with a high bulk milk SCC (>1,000,000 cells/mL) was investigated in October 2019 by the staff of the Faculty of Veterinary Medicine, Chiang Mai University. The results showed that *S. uberis* IMI was the dominant problem in this herd. A series of longitudinal studies were performed to monitor the dynamics of *S. uberis* IMI in this herd. Quarter milk samples from all cows were aseptically collected once a week. In all obtained milk samples, bacteriological culture, and identification were performed according to NMC standards [29]. Briefly, 0.01 mL of milk samples were cultured on a quarter of a 5% bovine blood agar plate and incubated for up to 24 h at 37°C. Colony morphology and biochemical tests were used for initial bacterial identification. Then, the identified *S. uberis* isolates were confirmed by PCR amplification [30].

The results of bacterial identification were used to determine whether *S. uberis* IMI could be classified as either a transient or persistent episode. Transient infections were defined as IMI with a duration of less than one week, whereas IMI with an infection duration of 8 weeks or longer was defined as a persistent infection [3, 31, 32]. Therefore, a transient *S. uberis* strain with a 1-week duration was selected from this herd, while a persistent *S. uberis* strain was obtained from our previous study [5]. The represented persistent strain was the dominant strain in the herd and was presumably contagiously transmitted in the herd and caused persistent intramammary infections for more than 10 months, confirmed by a similar PFGE pattern. The whole genome shotgun sequences of both transient [11] and persistent [5] *S. uberis* strains have been deposited at GenBank under the Biosample SAMN30958928 and SAMN30958927, respectively. Both of these strains were used as stimulating pathogens for the immune function tests.

### Preparation of milk-isolated white blood cells (mWBCs)

Five quarters with high SCC (SCC > 500,000 cells/ml) and negative bacterial results were sampled once during the study to obtain cells for immune tests. Thirty milliliters of the milk sample from each quarter were collected and immediately transported to the laboratory for mWBCs isolation. Briefly, quarter milk was centrifuged at 1,000 ×g for 10 min (Allegra X-15R Centrifuge, Beckman Coulter, Brea, CA, USA), and then the cream and whey layer was discarded. The remaining cell pellet was washed twice with Hank's balanced salt solution (HBSS, Sigma–Aldrich, St. Louis, MO, USA) and resuspended in cold RPMI-1640 medium (Gibco, Thermo Fisher Scientific, Waltham, MA, USA) supplemented with 1% heat-inactivated FBS (Gibco). Thereafter, the viability of retrieved mWBCs was assessed by trypan blue dye exclusion. The concentrations of mWBCs ranged between 1.2 and $7.6 \times 10^6$ cells/mL with $\geq 95\%$ viability. The differential cell counts of milk were determined from cytospin preparation slides. At least 500 cells from 5 random fields were manually counted and characterized mainly as neutrophils (60–65%), followed by macrophages. Finally, the mWBCs density was adjusted to approximately $1 \times 10^6$ cells/mL. The isolated mWBCs were randomly allocated into three cell populations, including an unstimulated population (HBSS) and populations stimulated with either the transient or persistent strain of *S. uberis*.

## Bacterial preparation for the immune function test

The selected bacterial isolates were recovered from stock (−80°C in brain heart infusion, 20% glycerol) in tryptic soy broth (TSB, HIMEDIA, Mumbai, India) and incubated aerobically at 37°C overnight. After, the inoculums were cultured on 5% bovine blood agar (HIMEDIA, Mumbai, India, with 5% washed bovine erythrocytes) at 37°C for 24 h. The bacterial inoculum was adjusted to approximately $10^8$ CFU/ml. For migration, bacterial killing, and NETosis assays, *S. uberis* bacteria were opsonized with 10% heat-inactivated normal bovine serum for 20 min at 37°C prior to use in the experiments. For phagocytosis and ROS assays, fluorescent *S. uberis* was prepared according to Chuammitri, Amphaiphan and Nojit [33] and Chuammitri, Srikok, Saipinta and Boonyayatra [34] with modifications. In brief, bacterial pellets ($10^8$ CFU/mL) were stained with $H_2$DCFDA (Invitrogen, Thermo Fisher Scientific, Waltham, MA, USA) or propidium iodide (Sigma–Aldrich) for ROS and phagocytosis assays, respectively. The fluorescently labeled *S. uberis* was resuspended, the inoculum was adjusted to $10^8$ CFU/mL with HBSS, and the samples were stored at 4°C until use.

## Immune function of milk-isolated white blood cells (mWBCs)

**Transwell *In Vitro* migration assay.** Briefly, 600 μL of RPMI-1640 or live *S. uberis* ($3 \times 10^5$ bacteria) were separately seeded into the lower chambers of a Transwell device. Then, a polycarbonate membrane Transwell insert (8 μm pore size, Corning, Corning, NY, USA) was placed over the lower chambers. One hundred microliters of mWBCs were added to Transwell inserts or upper chamber and incubated for 60 minutes at 37°C. After incubation, the liquid portion of the lower chamber containing migrated cells was collected for further analysis. Migrated cells were counted using forward scatter (FSC) and side scatter (SSC) in a flow cytometer according to a previously reported method [35].

**Phagocytosis assay.** The phagocytosis assay was performed as previously described with modifications [33]. Treated mWBCs ($3 \times 10^5$ cells) were mixed with opsonized fluorescently labeled transient *S. uberis* (MOI of 10) in duplicate in a 96-well plate. Labeling of persistent *S. uberis* strains was not successful (see discussion). The cell mixture was centrifuged at 1200 rpm for 3 min and incubated at 37°C and 5% $CO_2$ for 45 min. Data acquisitions (10,000 events) were performed by a CyAn ADP High-Performance Flow Cytometer (Beckman Coulter) with red laser (638 nm). Data were analyzed by FlowJo 10 (Treestar, Ashland, OR, USA) [36].

**Intracellular reactive oxygen species (ROS) assay.** To measure ROS production, mWBCs were activated to produce ROS with *S. uberis* (MOI of 10) in PBS with $Ca^{2+}$/$Mg^{2+}$ and then the cells were incubated for 30 min at 37°C with 5% $CO_2$. Subsequently, 10 μM $H_2$DCF-DA (Thermo Fisher Scientific, Waltham, MA, USA) was loaded into each well to stain intracellular $H_2O_2$ and the samples were incubated in the dark for 15 min [33]. Data acquisition and analysis were performed as stated in the phagocytosis assay with green laser (561 nm).

**Bacterial killing assay.** To measure the percentage of viable bacteria, a semiquantitative 2,5-diphenyl-2H-tetrazolium bromide (MTT) assay was used [33]. For this assay, mWBCs ($3 \times 10^5$ cells) were loaded into duplicate wells of a 96-well plate, and opsonized *S. uberis* was added to the MOI at 10 bacteria to 1 mWBC. Subsequently, the plate was centrifuged (1200 rpm, 3 min) and placed in an incubator for 45 min. After incubation, the plate was again centrifuged to remove non-ingested bacteria. Hypotonic solution (di$H_2O$) was used to release internalized bacteria from lysed neutrophils. Then, TSB mixed with 2 mg/ml MTT was added to all wells and the plates were incubated for 90 min at 37°C. Dimethyl sulfoxide (DMSO) was added to solubilize the MTT-insoluble formazan to colored crystals. Colorimetric detection was performed at a wavelength of 570 nm. The percentage of killing was calculated by

substituting the measured OD values into the following formula:

$$\% \text{ of killing} = 100 - [(\text{OD}_{\text{sample}} - \text{OD}_{\text{MTT}}) \times 100]$$

**Neutrophil extracellular trap (NET) assay.** The mWBCs ($3 \times 10^5$ cells) were seeded in a 96-well flat plate in duplicate. Live *S. uberis* ($3 \times 10^6$ bacteria) was added to all wells. Subsequently, HBSS-CM was added and the samples were incubated at 37˚C and 5% $CO_2$ for 150 min. The plate was centrifuged at 1200 rpm for 3 min and the samples were resuspended in cold RPMI-1640. The supernatant containing extracellular DNA was transferred to new plates. NET-DNA was quantified using a fluorescent dye (Hoechst 33342, Thermo Fisher Scientific, Waltham, MA, USA) [35]. Fluorescence measurement of stained NETs was performed with a Synergy™ HT Multi-Detection Microplate Reader using an excitation wavelength of 360 nm and an emission wavelength of 470 nm. The relative fluorescence units (RFU) were recorded. NET structures were also confirmed by staining the NET structure with Hoechst 33342 (nuclei) and visualization with fluorescence microscopy, as described in a previous publication [35].

**Gene expression in milk-isolated white blood cells (mWBCs) by real-time RT-PCR.** To investigate the gene expression of mWBCs after encountering transient and persistent *S. uberis* (MOI of 10) for 2 h, RNA*later*-preserved RNA (Invitrogen) was extracted using RNAzol®RT (Sigma-Aldrich, St. Louis, MO, USA) following the manufacturer's instructions. RNA yields and concentrations were measured using a DU 730 nanoVette UV/VIS spectrophotometer (Beckman Coulter). RNA purity was assessed using the A260/A280 ratio, and only samples with a ratio greater than 1.8 were selected for analysis. Twenty nanograms of total RNA was used for cDNA synthesis with a cDNA Synthesis Kit (Bioline, Taunton, MA, USA). To determine mRNA expression, 100 nanograms of cDNA was analyzed using real-time RT-PCR with a SensiFAST SYBR Hi-ROX Kit (Bioline, Taunton, MA, USA). The oligonucleotide primers used in this study are presented in Table 1. These primers included targets for Toll-like receptor genes (TLR1, TLR2, and TLR6), proinflammatory cytokines (TNF-α, IL-1β, and IL-8), gene-associated phagocytosis (RAC-1 and LAMP-1) and gene-associated ROS (SOD-1 and NOX). Measurements of the levels of gene expression were performed on an Applied Biosystems 7300 real-time PCR system equipped with SDS software v1.4 (Life Technologies)

**Table 1. Details of real-time RT-PCR primer sequences.**

| Genes[a] | Forward primer sequence | Reverse primer sequence | Amplicon size (bp) | References |
|---|---|---|---|---|
| *TLR1* | ACTTTGTTGCTGGCAAGAGC | TTCGCTCTGGACAAAGTTGG | 101 | This study |
| *TLR2* | TGCTGCCATTCTGATTCTGC | AACCAAAACCCTTCCTGCTG | 138 | This study |
| *TLR6* | AACTTTGTTGCCGGCAAGAG | ACTCGCTCTGGACAAAGTTG | 103 | This study |
| *TNF-α* | AGCACCAAAAGCATGATCCG | TTTGAACCAGAGGGCTGTTG | 226 | [34] |
| *IL1β* | ACAAAAGCTTCAGGCAGGTG | AGCACCAGGGATTTTTGCTC | 226 | [34] |
| *CXCL8* | TCTCTGCAGCTCTGTGTGAAG | TTCCTTGGGGTTTAGGCAGAC | 209 | [34] |
| *RAC1* | TGCCAATGTCATGGTGGATG | ACAATGGTGTCGCACTTCAG | 193 | [38] |
| *LAMP1* | ACAACGTTTCTGGCAGCAAC | GGTCTTGTTGGGGTTGACATTG | 125 | [38] |
| *SOD1* | GCTGACAAAAACGGTGTTGC | TCATTTCCACCTCTGCCCAAG | 131 | This study |
| *NOX1* | TGTCTGGGGTCAAACAGAAGAG | TTCAAATTGGGGAGGCTTGC | 112 | This study |
| *ACTB* | TGCGGCATTCACGAAACTAC | AGGGCAGTGATCTCTTTCTGC | 146 | [34] |

[a]Genes: *TLR*: Toll-like receptor, *TNF*: Tumor necrosis factor, *IL*: Interleukin, *RAC*: Rac Family Small GTPase, *LAMP*: Lysosomal associated membrane protein, *SOD*: Superoxide dismutase, *NOX*: NADPH Oxidase, *ACTB*: Actin Beta

Subsequently, specificity was confirmed by dissociation curve analysis ($T_m$). Actin Beta (*ACTB*) was used as internal control for gene expression normalization. The averages of intra-coefficient of variation (CV) values of ACTB were 1.588%, 0.977%, and 0.824% for unstimulated, stimulated with transient, and persistent *S. uberis*, respectively, which are below 10% indicating that ACTB was a reliable reference gene. The expression levels (fold difference) were reported using the $2^{-\Delta\Delta C_T}$ method [37].

## Statistical analysis

Statistical analysis was performed using the SAS University Edition (SAS Institute Inc., Cary, NC). When necessary, data were log10 transformed to maintain the assumption of normality. Statistical significance was assigned at a $P < 0.05$. The differences in immune responses after restimulation with two different strains of *S. uberis* were determined using repeated analysis of variance (ANOVA) to compare the means of the three treatment groups. The least-square means with Tukey's HSD adjustment was used to compare the groups, and statistical significance was assigned at a $P < 0.05$. Information obtained from the statistical analysis was presented as graphs generated by GraphPad Prism version 6 (GraphPad Software, San Diego, CA). Gene expression patterns are presented in the form of boxplots and heatmaps.

## Results

Flow cytometry results measuring mWBCs migration toward live *S. uberis* strains or HBSS are shown in Fig 1. The density of the dots indicates the number of migrated cells. Among the 3 stimulants, the highest density of mWBCs was found when migrating toward the transient *S. uberis*, as depicted in Fig 1B. Significant differences in the number of migrated mWBCs were found in response to transient *S. uberis* (6365.2 ± 951 cells, p < 0.05) compared to persistent *S. uberis* or HBSS. No significant difference was found in the number of migrated cells toward persistent *S. uberis* compared to HBSS (714 ± 41.1 and 310 ± 117.0 cells, respectively). The in vitro phagocytosis of PI-labeled transient *S. uberis* by mWBCs is depicted in Fig 2. Encounters with transient *S. uberis* resulted in significantly reduced phagocytosis of stimulated mWBCs (22.2 ± 2.2) compared to that with HBSS (36.4 ± 3.7; p < 0.05, Fig 2B). However, as shown in Fig 2C, PI-labeled *S. uberis* was phagocytosed by mWBCs.

The histogram presenting the MFI of ROS production against different stimuli is depicted in Fig 3B. Flow cytometry data showed that mWBCs stimulated with transient *S. uberis* produced significantly larger amounts of ROS (1657.2 ± 138) than those stimulated with persistent *S. uberis* and HBSS (587 ± 76.5 and 555 ± 23.7, respectively, p < 0.0001). As depicted in Fig 3C, fluorescently stained cells indicated the intracellular ROS production of mWBCs stimulated by *S. uberis* and $H_2O_2$. These observations confirmed that *S. uberis* cells stimulated the production of intracellular ROS at the same level as $H_2O_2$, which served as a positive stimulus control. With regard to the intracellular antibacterial mechanisms, the percentage of killed bacteria was significantly higher in the case of the transient strain compared to the persistent strain. The visible difference in the color of MTT revealed that mWBCs killed *S. uberis* (Fig 4). High amounts of dead transient *S. uberis* were shown by a less dense purple color. The mWBCs showed a significant difference in the amount of killed bacteria between groups, where the percentage of the bacterial killing of mWBCs for the transient strain (91.75 ± 0.9%) was significantly higher than that for the persistent strain (54.43 ± 4.9%) at p < 0.001.

The indirect killing of extracellular *S. uberis* by NETs was assessed using a fluorescence plate reader. The significant difference in NET release after stimulation with a transient and persistent strain of *S. uberis* was 1.17 ± 0.1 and 1.03 ± 0.1, respectively (p = 0.001, Fig 5A). As shown in Fig 5B, the extracellular structures of NETs were also visually confirmed by cytospin preparation

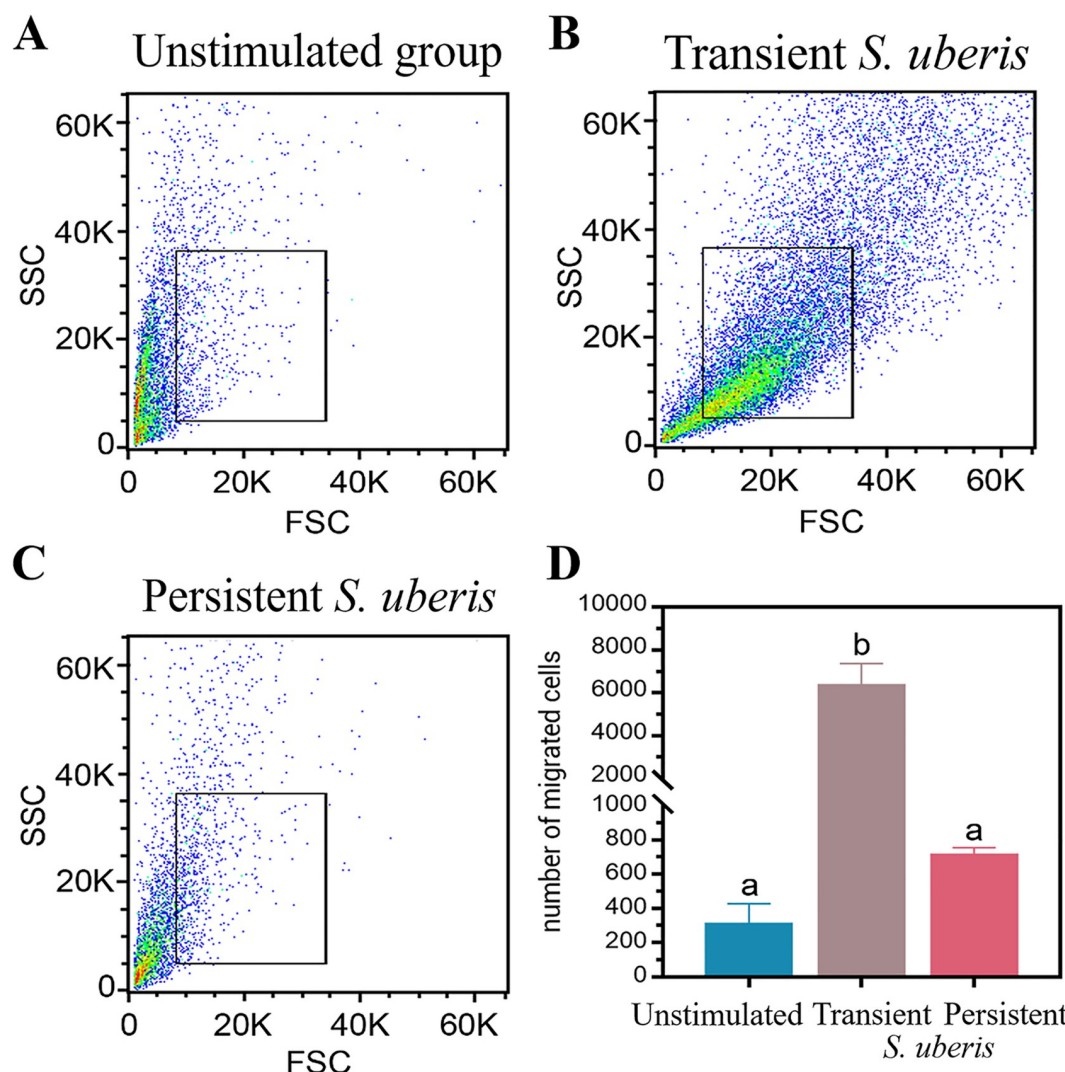

**Fig 1. Flow cytometry results of migrated milk-isolated white blood cells.** (A-C) Dot plots show the quantity of migrated cells following exposure to various stimulants. (D) Bar graphs show the number of migrated cells toward HBSS and transient and persistent *S. uberis*. Data are the mean ± SEM of three independent experiments (n = 5 each treatment), $^{a,b}$ p < 0.05.

slides after DipQuick staining. The NET structures stimulated by different stimulants showed different DNA-stained structures that protruded and lengthened away from the *S. uberis*-stimulated cells. The protruded DNA of mWBCs stimulated by persistent *S. uberis* showed a NET-like structure that was comparable to the positive control using PMA, while a hook-like NET structure was found in following exposure to transient *S. uberis*. Gene Expression in mWBCs.

The differential mRNA expression of *S. uberis*-activated mWBCs after stimulation for 2 h is shown in Fig 6. As shown in Fig 6B, our findings indicated that most genes were significantly upregulated in mWBCs activated with transient *S. uberis*, except genes involved in phagocytosis (RAC) and inflammatory cytokines (IL-8). For the recognition genes, significantly higher expression of the TLR1 and TLR2 genes was observed in transient *S. uberis* compared to persistent *S. uberis* (p < 0.05). The genes were elevated 1.737-fold and 1.339-fold for TLR1 and TLR2 upon transient *S. uberis* stimulation compared to unstimulated group, respectively. For proinflammatory cytokine genes, TNF-α expression in both *S. uberis* stimulations was significantly higher than that in the HBSS control group, whereas the expression of IL-8 was

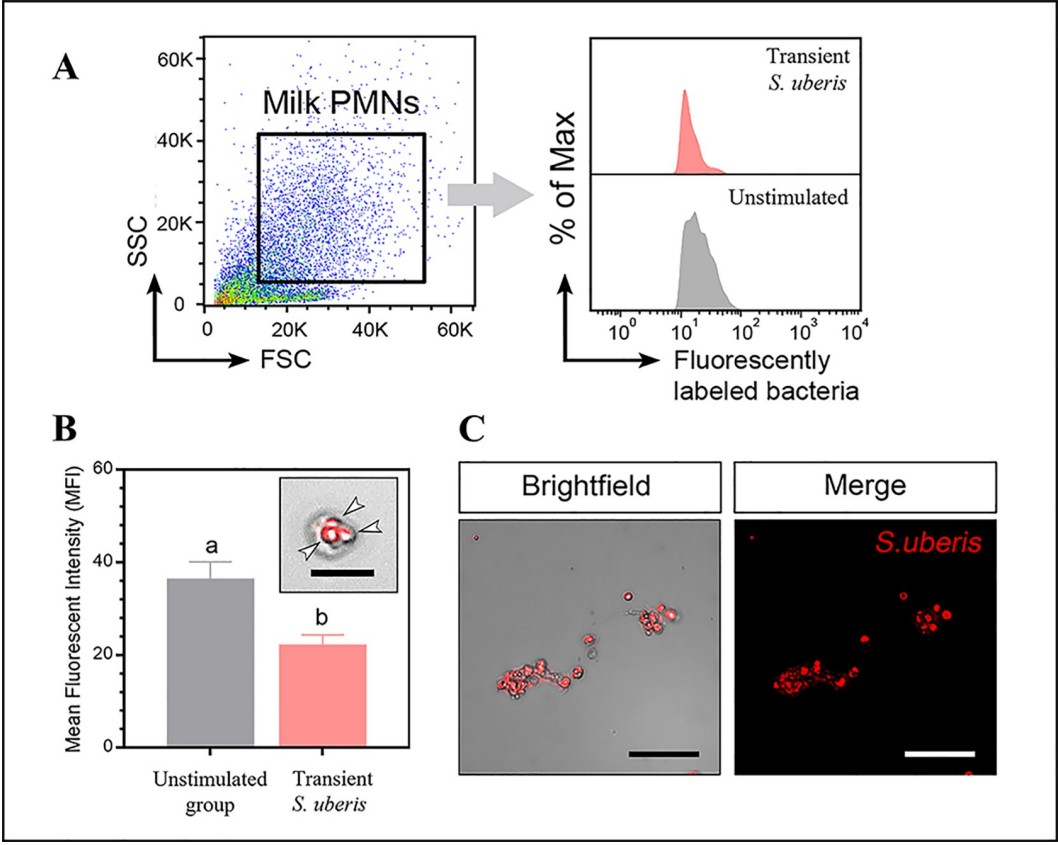

**Fig 2. Flow cytometry results of phagocytosis in milk-isolated white blood cells.** (A) Dot plots and histograms show phagocytosis of fluorescently labeled transient *S. uberis* in stimulated milk cells. (B) Bar graphs show the mean fluorescence intensity (MFI), an indicator of the percentage of phagocytosis in response to either HBSS or transient *S. uberis*. Data shown are the mean ± SEM of three independent experiments (n = 5 each treatment), [a,b] p < 0.05. The arrowhead points to phagocytosing milk leukocytes, 200× magnification. (C) Representative images of phagocytic cells as visualized by PI-stained transient *S. uberis*.

significantly lower in both the transient and persistent *S. uberis* groups than the HBSS group. Additionally, the expression of NOX-1, a gene related to oxidase production, was significantly higher following exposure with persistent *S. uberis* (2.695-fold) than with transient *S. uberis* (1.299-fold, p < 0.05). No significant difference was found in the expression of the genes RAC-1, LAMP-1, SOD-1, and IL-1β.

To summarize the expression patterns described above, a heatmap was generated using the qRT–PCR results, and it depicted a z score, shown as a color scale, of relative mRNA abundance after the stimulation of the mWBCs with either transient or persistent *S. uberis*. This heatmap of gene expression that indicates the expression of genes is presented in Fig 6A, where a blue color represents low expression, and a red color represents high expression. The overall mRNA expression was low to moderate after stimulation. However, TNF-α, RAC, and IL-8 were highly expressed (red color indication), and the highest mRNA levels were found in transient, persistent, and unstimulated groups of mWBCs, respectively.

## Discussion

This study aimed to differentiate the in vitro immune response of milk WBCs against two *S. uberis* strains that induce different durations of IMI and with or without a spontaneous cure.

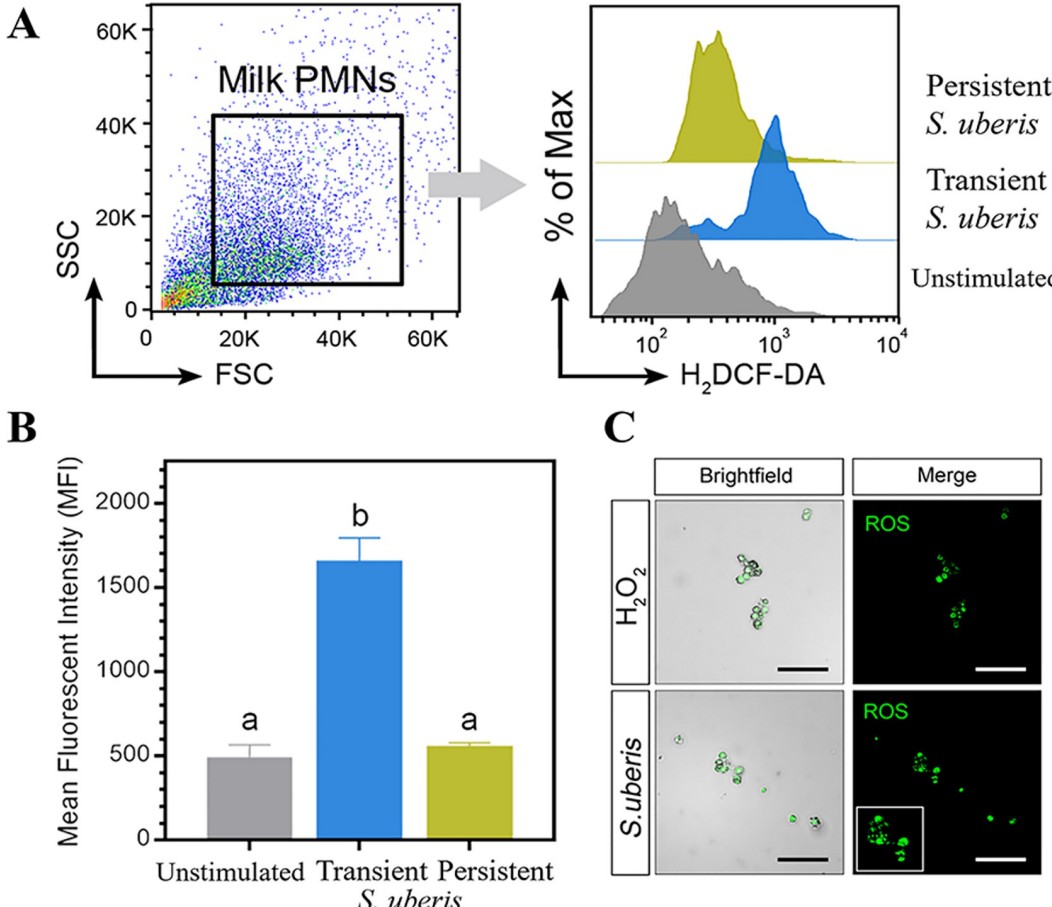

**Fig 3. Flow cytometry results of stimulated milk-isolated white blood cells.** (A) Dot plots and histograms show the ROS production of treated cells. (B) Bar graphs show the MFI of ROS production induced by HBSS and transient or persistent *S. uberis*. Data are the mean ± SEM of three independent experiments (n = 5 each treatment), [a,b] $p < 0.05$. (C) Characterization of stimulated mWBCs for ROS molecules stained with $H_2$DCFDA for nuclear materials.

These two strains were defined as giving rise to either transient (1 week in spontaneous cure) or persistent IMI (>2 months without an observed spontaneous cure and confirmed persistence by PFGE pattern). The persistent strain was referred to as PFGE type A by Leelahapongsathon, Schukken, Srithanasuwan and Suriyasathaporn [5]. According to the marked differences in effector functions, gene expression, especially that of TLR and TNF-α genes, and genomic data, significant phenotypic and genotypic differences between exposure to transient and persistent *S. uberis* strains were observed. We hypothesize that these observations may be generalized, meaning that the two strains used here represent strains in each IMI group. Since only two strains were used in this study, the results should be interpreted with care. However, the strains were obtained under field conditions and met quite extreme IMI duration criteria. Moreover, both strains utilized in this study were isolated from herds that took part in a mastitis investigation, and *S. uberis* was found to be the dominant pathogen. Due to the continuing poor milk quality management in farms, high rates of transient and persistent IMI results in continuous dominance of *S. uberis* infections. However, despite the limited number of tested strains, the results of inter-strain differences are consistent with previous studies [26, 39] and revealed specific characteristics of either transient or persistent *S. uberis*.

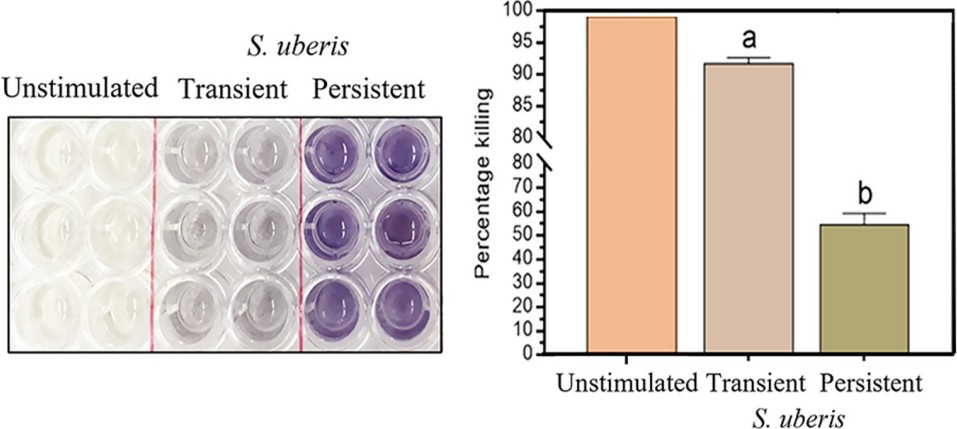

**Fig 4. Results from the MTT assay after treatment with transient or persistent *Streptococcus uberis*.** Representative images (left) and bar graphs show the percentage of live *S. uberis* in milk mWBCs in the unstimulated group and the group treated with transient and persistent *S. uberis*. Data are the mean ± SEM of three independent experiments (n = 5 each treatment), [a,b] p < 0.05 (Right).

In general, *S. uberis* appears as small (1–3 mm in diameter) translucent colonies with a moist, convex, and dense center. *S. uberis* has several virulence factors, including SUAM, which is involved in adherence to, internalization into, and persistence of *S. uberis* in bovine mammary epithelial cells [40]. In our previous study, we found that both of these *S. uberis* strains showed *S. uberis* adhesive molecule (SUAM) [41] as one of their virulence factor genes

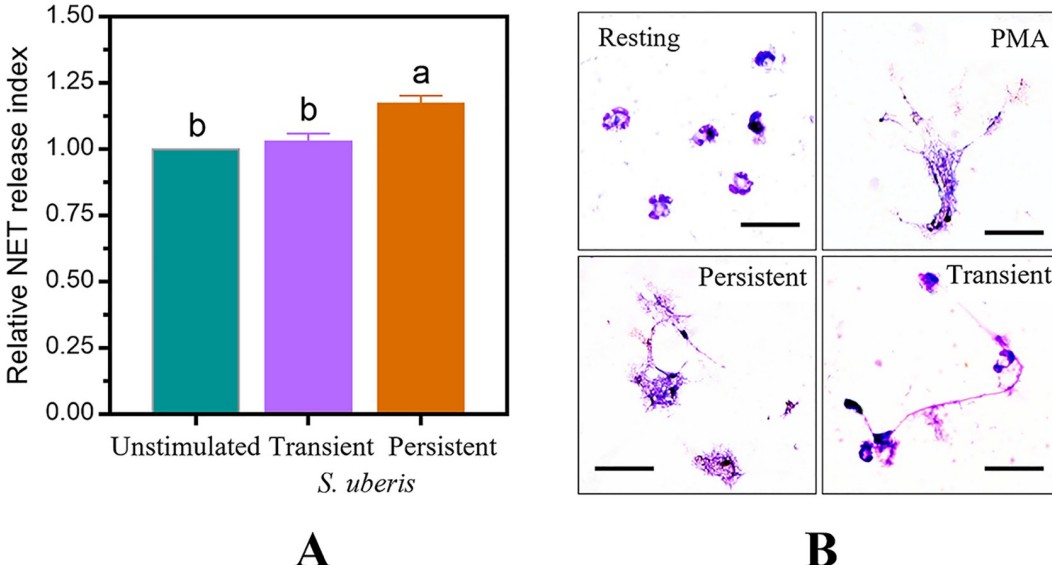

**Fig 5. NET production and release after treatment with transient or persistent *Streptococcus uberis*.** The bar graphs show the quantity of NETs in the unstimulated (HBSS) and stimulated with either transient or persistent *S. uberis* groups. The average data of the relative NET index over unstimulated group ± SEM of three independent experiments (n = 5 per treatment); [a,b] p < 0.05. (B) Representative images from cytospin preparation slides depict NET structures at the resting stage and following stimulation by phorbol-12-myristate-13-acetate (PMA; positive control), transient and persistent *S. uberis* as visualized by DipQuick staining, 20x magnification.

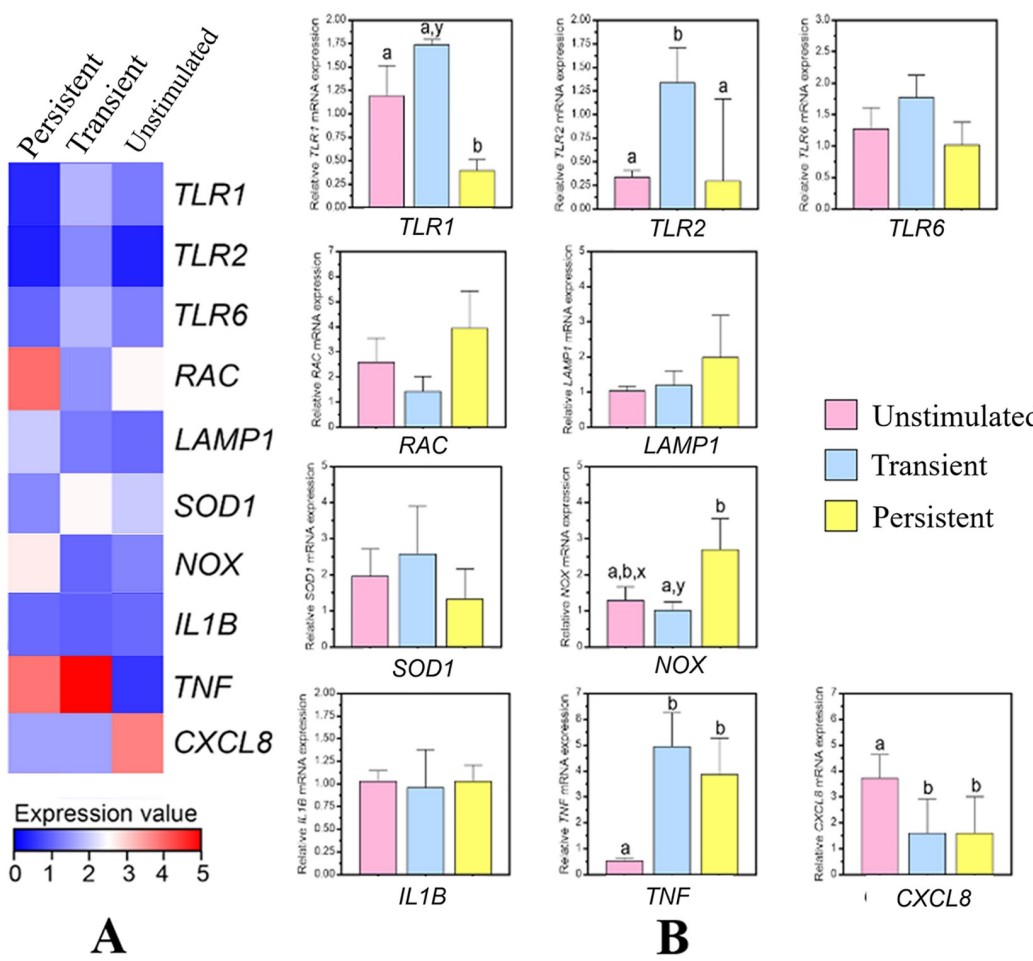

**Fig 6. Real-time RT–PCR analyses of milk-isolated white blood cells genes involved in response to *S. uberis*.** The heatmap (A) shows relative *TLR-1*, *TLR-2*, *TLR-6*, *RAC*, *LAMP-1*, *SOD-1*, *NOX-1*, *IL-1β*, *TNF-a*, and *IL-8* expression after normalization to 3-actin expression in the unstimulated, transient or persistent *S. uberis* groups. (B) The bar graph shows the results represented as the mean ± SEM (n = 5 each treatment), one-way ANOVA, [a,b] p < 0.05, [x,y] p < 0.1.

[11]. The persistent strain used in this study expressed the *hasABC* locus responsible for the production of hyaluronic acid capsules [11]. These physiological differences may relate to differences in the in vitro immune response; for example, bacterial resistance to phagocytosis and susceptibility to antimicrobials [42, 43]. This phenotypic heterogeneity in biofilm production exhibited by *S. uberis* could at least partly explain why this bacterium has the ability to adapt to different niches and survive under diverse and stressful conditions [44].

The presence of genetic markers such as the *hasABC* locus may eventually lead to a biomarker that may be used to identify high-risk strains for persistent *S. uberis* IMI. Such genetic tools would be very helpful in developing selective control programs for either transient or persistent *S. uberis* strains [6]. An abundance of persistent strains in a herd would point to control programs closely related to contagious mastitis control and focus on reducing duration and transmission. However, an abundance of transient strains in a herd would point to control programs for environmental mastitis with a focus on infection reduction and reducing the severity of the inflammatory response. A much larger collection of transient and persistent isolates would be essential to identify such bacteriological biomarkers.

Milk WBCs were used instead of blood WBCs to closely imitate the udder defense mechanism after bacterial invasion. Differences in immune functions were observed between activated polymorphonuclear leukocytes (PMNs), such as peritoneal exudate PMNs, and nonactivated bone marrow PMNs, whereby the activated PMNs are fully primed when isolated [45]. Thus, activated PMNs were more appropriate for investigations into the pathophysiological functions of PMNs at sites of inflammation [45]. The mWBCs used in this study were obtained from milk samples without any identification of bacteria, indicating no infection in the source quarters. However, the high somatic cell counts observed in these milk samples, ranging between 1.2 and 7.6 x 106 cells/ml, which is in the range of milk with subclinical mastitis [46], might be related to their recent recovery from mastitis. Additionally, the majority of cells found in our study were neutrophils (approximately 60%), indicating a continuing inflammatory response due to neutrophil transmigration to the inflamed mammary gland [17]. In inflamed quarters, the number of neutrophils in the immediate response to acute intramammary infection may reach 90%. The neutrophils in the samples in this study might confirm the status of mWBCs in this study as activated. Therefore, the results of this study should be interpreted carefully, and full interpretation may be limited to the immune function of activated mWBCs.

The quality and quantity of early neutrophil recruitment are critical for bacterial clearance and host survival [47, 48]. The mWBCs in this study were tested for their response related to both the quality and quantity of recruitment. The mWBCs in this study showed expression of some Toll-like receptor genes (TLR-1 and TLR-2) and a proinflammatory cytokine gene (TNF-α). This expression was significantly higher when stimulated with transient *S. uberis* compared to the expression when stimulated with either unstimulated or persistent *S. uberis* (Fig 6). Our flow cytometry results also indicated that the number of migrated mWBCs after stimulation with transient *S. uberis* was higher than that in unstimulated cells or cells stimulated with persistent *S. uberis* (Fig 1). When TLR activation is triggered, it initiates significant cellular processes such as reactive oxygen species (ROS) generation, production of cytokines, and enhanced survival. These processes, when their signaling is dysregulated, can contribute to the development of chronic inflammation and its associated pathogenesis [49]. Less TLR-1 and TLR-2 expression due to persistent *S. uberis* might cause persistent intramammary infection. Recognition of conserved bacteria by TLRs leads to a variety of signals related to immune function, including proinflammatory cytokine production, costimulatory molecule upregulation, antimicrobial peptide secretion, and phagosome maturation [50, 51]. Expression of the TNF-α gene promotes neutrophil migration [52]. Fewer migrated mWBCs, as a result of stimulation with persistent *S. uberis*, support the results of previous studies showing that a lower or delayed influx of PMNs may lead to persistent infection [31, 53]. The lack of a significant difference in migrated mWBCs between unstimulated cells and cells stimulated with persistent *S. uberis* may be related to the lower expression of TLR1 and the higher TNF-α when stimulated with persistent *S. uberis*. Different responses between transient and persistent *S. uberis* can be explained by the theory of bacterial virulence factors [54]. All *S. uberis* isolated from bovine IMI may adhere to and invade mammary epithelial cells, which requires intact microfilaments and the de novo eukaryotic protein synthesis that is required for bacterial invasion [55]. In our recent study, different patterns of gene expression of 6 virulence factors were found when comparing responses to transient and persistent *S. uberis* strains [11], where the transient strain had reduced expression of virulence factors.

While freshly migrated PMNs are active phagocytes, continued exposure of PMNs to inhibitory factors in milk, such as fat globules and casein, as our mWBCs before collection, might lead to altered PMN morphology and reduced phagocytosis [17]. In comparison to our selected genes related to phagocytosis, neither RAC nor LAMP-1 expression was different

among cells stimulated with HBSS, transient *S. uberis*, or persistent *S. uberis* (Fig 6). The phagocytosis measured using flow cytometry in this study was firstly designed to compare the activity between fluorescently labeled transient and persistent S. uberis stimulated mWBCs. Unfortunately, fluorescent labeling of persistent *S. uberis* could not be performed, and therefore, no phagocytosis test of mWBCs using persistent *S. uberis* was performed. The inability to label persistent *S. uberis* might be related to its biofilm formation and production of hyaluronic acid capsules [11], which may interfere with the PI-based staining of encapsulated bacteria [56]. Exposure to *S. uberis* with hyaluronic acid capsule expression leads to reduced phagocytosis [42], indicating that stimulation with persistent *S. uberis* might lead to less phagocytosis than stimulation with transient *S. uberis*. The finding of the lower phagocytosis activity of mWBCs stimulated with *S. uberis* than mWBCs stimulated with HBSS (Fig 2) might be due to the fluorescent response of mWBC after stimulation. The fluorescence detected in WBC stimulated with HBSS was attributed to autofluorescence with more intensity in the activated PMNs [57] as the mWBC used in this study. The less fluorescence mWBC stimulated with transient *S. uberis* might be caused by blocking the emission of fluorescent light or quenching the fluorescence of *S. uberis* within mWBCs by absorbing the excitation light, resulting in a weaker fluorescent intensity as depicted in Fig 2B.

ROS production, i.e., oxidative burst, is a powerful antimicrobial weapon and a major component of the innate immune defense killing mechanism against bacterial and fungal infections [58, 59]. In this study, the killing efficacy of mWBCs after culture with each *S. uberis* strain showed that mWBCs showed a higher percentage killing of transient *S. uberis* compared to persistent *S. uberis*. In support of the results from flow cytometry, in mWBCs, ROS production was significantly higher after stimulation with transient *S. uberis* than in unstimulated cells or cells stimulated with persistent *S. uberis* (Fig 3). However, the expression of NOX-1 in mWBCs stimulated with persistent *S. uberis* was significantly higher than that in cells stimulated with transient *S. uberis*, while no difference in SOD-1 expression was observed. NOX enzymes, from a family of NADPH oxidases, produce superoxide. Subsequently, superoxide dismutase enzymes (SOD) dismutase this superoxide into $H_2O_2$ [60] to prevent the overgrowth of commensal bacteria [61]. The lower expression of genes related to ROS production might indicate that the killing efficacy after phagocytosis of persistent SU was lower. The killing efficacy of the whole process, from adhesion to ROS production and ROS labeling of mWBCs, was higher when cells were stimulated with transient *S. uberis*. This is likely due to increased migration, higher cytokine production, and increased phagocytosis after stimulation with transient *S. uberis*, as previously discussed.

Failure of phagocytosis may result in the formation of neutrophil extracellular traps (NETs) with proinflammatory effects [62]. As revealed by this in vitro study, the persistent *S. uberis* strain induced more NETs with different structures compared to that of the transient strain (Fig 5). This is consistent with other studies that reported a high amount of NET release in persistent infection [63, 64]. Swain, Kushwah, Kaur and Dang [65] described a positive correlation between delayed neutrophil apoptosis, the persistence of neutrophils at the site of infection, and the formation of NETs. Although results from previous studies suggested that the formation of NETs is pathogen-specific [66, 67], the mechanism of strain-specific regulation of NET release has rarely been studied.

Therefore, the results from this current study, similar to that of our previous study, indicate that transient and persistent strains of *S. uberis* may elicit different immune responses by mWBCs. One of the contributing factors may be the virulence factors of bacteria, such as hyaluronic acid capsules. Knowledge of strain-specific interactions of transient and persistent *S. uberis* with the host immune response will provide insight and opportunities to ameliorate the severity and duration of intramammary bacterial infections and may eventually lead to better

mastitis control programs and interventions. However, the immune response may be affected by ability to colonization due to different virulence of bacteria or interfered by milk components, further investigations, including *in vivo* studies, should be pursued to better understand the strain-specific response and determine factors that influence the interaction between local immunity and strain-specific *S. uberis* IMI in the udder microenvironment.

## Supporting information

**S1 File. Data of CMT score and somatic cell count, RNA concentration and purity values of the samples used in the study, and primer and gene efficiencies.**
(PDF)

**S2 File. Illustration of mWBC assay, strain genomic data, and the primer efficiencies for the genes.**
(PDF)

## Acknowledgments

The assistance provided by our colleagues, Ms. Nadruedee Leenarach, Mr. Jakkrawut Srachum, and Ms. Laorat Tata, Faculty of Veterinary Medicine and Research Center of Producing and Development of Products and Innovations for Animal Health and Production, Chiang Mai University, Chiang Mai, Thailand, was greatly appreciated.

## Author Contributions

**Conceptualization:** Phongsakorn Chuammitri, Witaya Suriyasathaporn.

**Formal analysis:** Witaya Suriyasathaporn.

**Investigation:** Anyaphat Srithanasuwan.

**Methodology:** Anyaphat Srithanasuwan, Noppason Pangprasit, Phongsakorn Chuammitri.

**Supervision:** Ynte H. Schukken, Phongsakorn Chuammitri, Witaya Suriyasathaporn.

**Validation:** Ynte H. Schukken, Noppason Pangprasit, Witaya Suriyasathaporn.

**Writing – original draft:** Anyaphat Srithanasuwan, Witaya Suriyasathaporn.

**Writing – review & editing:** Ynte H. Schukken, Phongsakorn Chuammitri, Witaya Suriyasathaporn.

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
