## [Decision Letter · Decision Letter 0]

14 Jun 2023

PONE-D-23-14210Different cellular and molecular responses of bovine milk phagocytes to persistent and transient strains of *Streptococcus uberis* causing mastitis.PLOS ONE

Dear Dr. Suriyasathaporn,

Thank you for submitting your manuscript to PLOS ONE. After careful consideration, we feel that it has merit but does not fully meet PLOS ONE’s publication criteria as it currently stands. Therefore, we invite you to submit a revised version of the manuscript that addresses the points raised during the review process.

I also encourage you to revise the methods section : the protocol used for the migration assay should clarify what you mean by "stimulated mWBCs". The protocol for the isolation of mWBC is presented, the protocol for the preparation of bacteria is also clearly presented but, as mentioned by reviewer #2, the protocol for stimulation step is missing. As I understand it, for the migration assay, live. S.uberis are placed in the upper chamber and "stimulated" mWBC are placed in the lower chamber. Is it correct ? or are mWBC placed in the lower chamber and are stimulated by the release of some components from the bacteria present in the upper chamber ?

You should also clarify the meaning of "moderately high somatic cell count" and give a range of SCC.

How to make sure that mWBC are actually PMN as depicted in figure 1 ?

For the bacterial killing assay, could you clarify the multiplicity of infection : is it 1 bacteria per 10 mWBC or 10 bacteria for 1 mWBC ? When is the formazan added in your protocol ?

For RT-qPCR experiments, I concur with reviewer #2. You did not specify the reference genes you used to calculate fold changes. MIQE guidelines recommend the use of more than one reference gene.

L290: it is not correct that TLR1 are respectively increased 1.7 compared to stimulation by persistent strains: these fold changes are against unstimulated mWBCs. Please revise also the fold-change for TLR2.

We look forward to receiving your revised manuscript.

Kind regards,

Pierre Germon

Academic Editor

PLOS ONE

Journal Requirements:

Reviewers' comments:

Reviewer's Responses to Questions

**Comments to the Author**

1. Is the manuscript technically sound, and do the data support the conclusions?

Reviewer #1: No

Reviewer #2: Yes

2. Has the statistical analysis been performed appropriately and rigorously? 

Reviewer #1: Yes

Reviewer #2: Yes

3. Have the authors made all data underlying the findings in their manuscript fully available?

Reviewer #1: Yes

Reviewer #2: Yes

4. Is the manuscript presented in an intelligible fashion and written in standard English?

Reviewer #1: Yes

Reviewer #2: Yes

5. Review Comments to the Author

Reviewer #1: In this study, the authors sought to understand the differences in host phagocytes responding to S.uberis strains which either cause persistent or transient mastitis in cows. Overall, it seems like a sensible approach was taken to understand these important cells in pathenogensis.

However, it is not clear how these strains were chosen, there is a reference to pulse field gel electrophoresis in the abstract, but this technique is not mentioned again. It would also benefit the broad readership of plos one if the authors described more detail about the capsular/noncapsular strains, and why this is important.

I am pleased to see the shotgun sequences of the strains used deposited in GenBank, it is worth the authors briefly analysing these data. Since these strains appear to be from the same location, how similar are they? There is evidence that some aetiology for mastitis is due to the host, so it is worth understanding the genetic differences in the chosen challenge strains, so that the conclusions from the measures of host responses are clearly derived from identifiable differences in the challenge.

That said, beyond this, there are some important and sensible experiments performed here that will benefit the field.

That said, there are still some limitations here that need to be addressed:

Recent work has shown that in vivo, the immune response benefits colonisation (DOI: 10.3390/pathogens9120997), with at least one pathogenic strain triggering the immune response and seeming to benefit from this pathologically. This is corroborated here by the increased NETs in response to the persistent strain. Archer et al (2020) included a measure of Il-1B protein output - could the authors comment on this and how this fits with their measurements of levels of immune associated mRNAs by qPCR? RNAs for immune responses are highly regulated by their translation. It is worth measuring one of these proteins by western blot or ELISA to properly characterise the host response, for example, it is likely that rapid translation of CXCL8 would reduce its overall RNA levels, in which case the qPCR result is showing the opposite of the actual host response! Alternatively, this could be the result of difference in the immune population being challenged - Archer et al isolated macrophage while the present study isolated all milk white blood cells.

The bacterial cell killing assay used by the authors is MTT, which was originally developed and best established for eukaryotic cells (relying on mitochondrial proteins). It generally still works with other organisms, but it is worth touching on this nuance in the methods (even in the text I would say is ok). Further, many immune responses are inhibited in the presence of milk, particularly against S.uberis. It is worth performing this assay in the presence of milk.

Should these concerns be addressed, I think this would be a worthy contribution to the field of understanding S.uberis pathogenesis, a very curious bacteria indeed!

Reviewer #2: The authors conducted an experiment to assess the effects of S. uberis strains on milk WBC immune responses. The paper needs some additional clarification on the methods, information requested is listed below. I also think there is a major limitation to using milk WBC from cows with subclinical mastitis, which the authors have addressed in the discussion; however, this important piece of information should be added to the abstract as well, so readers can interpret the data carefully.

Major comments on methods:

Lines 114-115: How are you defining negative bacterial results? Zero cfu? Or less than 300 cfu/mL? How much milk did you plate? More information on bacteriological culture is needed here and in line 102.

Lines 191-207:

1) Please describe the experiment here. How long were the mWBC challenged with S. uberis, how much S. uberis per cell, etc.?

2) Please provide RNA purity and RNA integrity values.

3) Please provide primer efficiencies. If primer efficiencies do not fall within 90-110% range, then please use the efficiency/Pfaffl equation rather than the delta CT method.

4) The use of 1 internal control gene is generally not recommended anymore. It is highly recommended to use at least 2, if not 3+ control genes. Please prove that this single control gene is very stable. There are algorithms to determine the number of control genes needed such as geNORM as well as others that I suggest that the authors use.

5) Table 1 - define the acronyms.

Statistics: Were the p-values adjusted for multiple comparisons? Tukey adjustment? With 3 treatment groups, this would be highly recommended.

Results:

Line 268-269 This sentence is awkward and needs reworded.

Figure 6 heading - please provide sample size used for these analyses

Discussion:

The sentence is line 366-367 makes an excellent point. I think this point should be added to the abstract as well.

Line 403 - remove the word 'help'

6. PLOS authors have the option to publish the peer review history of their article (what does this mean?). If published, this will include your full peer review and any attached files.

Reviewer #1: No

Reviewer #2: No

---

## [Author Response · Author response to Decision Letter 0]

31 Jul 2023

Manuscript ID: PONE-D-23-14210

Title: Different cellular and molecular responses of bovine milk phagocytes to persistent and transient strains of Streptococcus uberis causing mastitis.

The authors appreciate all comments and suggestions from the reviewers and the editor, who helped improve the manuscript. We have carefully revised the manuscript according to the comments, and all explanations and corrections are presented below. We have made all of the changes requested by the reviewers, and we have answered their questions individually.

Two versions of the article were provided to the reviewers: a clean version (PONE-D-23-14210_R1) and a version with tracked changes (PONE-D-23-14210_R1_Track_Changes). This was done to help them evaluate the manuscript.

Editor's comments:

Revise the methods section.

1. The protocol used for the migration assay should clarify what you mean by "stimulated mWBCs."

Response 1: The meaning of "mWBC" in this study is the white blood cells (WBCs) isolated from milk that bacteria and chemoattractants have naturally stimulated in the infected quarter. This stimulation causes the migration of blood WBCs into the infected quarter, where they become mWBCs. As discussed in lines 370–375, we agree that the word "stimulated" in the phrase "stimulated mWBC" could confuse readers. 

Solution: we have removed the word "stimulated" from lines 158 and 190 of the revised manuscript.

2. The protocol for the isolation of mWBC is presented, the protocol for the preparation of bacteria is also clearly presented but, as mentioned by reviewer #2, the protocol for stimulation step is missing.

Response 2: As described in Response 1, the stimulation occurred naturally before milk collection; the word "stimulation" would confuse readers. 

Solution: We have removed the word "stimulated" from lines 158 and 190 of the revised manuscript.

3. For the migration assay, live. S.uberis are placed in the upper chamber, and "stimulated" mWBC are placed in the lower chamber. Is it correct? or are mWBC placed in the lower chamber and are stimulated by the release of some components from the bacteria present in the upper chamber?

Response 3: In our method, we seeded live S. uberis (3 × 105) into the lower chambers of a Transwell culture system. We then added mWBCs to the upper chamber (Transwell insert). This is illustrated in the Figure "Supplementary Figure S1", which shows the experimental setup from lines 155–161 of the manuscript.

4. You should also clarify the meaning of "moderately high somatic cell count" and give a range of SCC.

Response 4: The National Mastitis Council (NMC) defines abnormal SCC as a somatic cell count (SCC) of >200,000 cells/ml. Our manuscript uses the term "high SCC" to refer to SCC values exceeding 500,000 cells/ml. Our study's SCC range was from 628 to 9,953 × 103 cells/ml. We have attached the results of the SCC count in the supplementary table S1.

Solution: We have edited line 123 to reflect this change, replacing the phrase "moderately high SCC" with "high SCC (SCC > 500,000 cells/ml)".

5. How to make sure that mWBC is actually PMN ?

Response 5: A cytospin preparation slide was used to check the differential cells of mWBC in the manuscript. The results showed that neutrophils were the majority population, as shown in lines 132–135. We also provide an example of a cytospin that illustrates the isolated mWBCs (Supplementary Figure S2).

6. For the bacterial killing assay, could you clarify the multiplicity of infection: is it 1 bacteria per 10 mWBC or 10 bacteria for 1 mWBC? When is the formazan added in your protocol?

Response 6: The authors have checked and corrected the information. 

Solution: Lines 179–180 have been changed to read: "opsonized S. uberis was added to the MOI at 10 bacteria to 1 mWBC." Line 183: Added information on the formazan step as "TSB mixed with 2 mg/ml MTT".

7. For RT-qPCR experiments, I concur with reviewer #2. You did not specify the reference genes you used to calculate fold changes. MIQE guidelines recommend the use of more than one reference gene.

Response 7: Thank you for commenting on using reference genes in our RT-qPCR experiments. We apologize for not specifying the reference genes in the manuscript. Our experiments used one reference gene (ACTB or β-actin). The reference gene used to calculate fold changes was described in lines 217–218 in the revised manuscript by adding, "Actin Beta (ACTB) was used as the internal control for gene expression normalization." The β-actin is a well-characterized housekeeping gene known to be relatively stable in most tissues and cell types (Zeng et al., 2016). We have calculated the fold changes for all of the genes in our study using only one reference gene. 

We understand that the MIQE guidelines recommend using more than one reference gene. However, we believe that using one reference gene is sufficient for our study. We have conducted preliminary experiments to demonstrate that the expression levels of the reference gene and other genes have high PCR efficiency. Please refer to Response No. 16 for further details.

Reference: Zeng J., Liu S., Zhao Y., Tan X., Aljohi HA., Liu W., Hu S. (2016). Identification and analysis of housekeeping and tissue-specific genes based on RNA-seq data sets across 15 mouse tissues. Gene, 560-70. DOI: 10.1016/j.gene.2015.11.003.

8. L290: it is not correct that TLR1 are respectively increased 1.7 compared to stimulation by persistent strains: these fold changes are against unstimulated mWBCs. Please revise also the fold-change for TLR2.

Response 8: The authors acknowledged that the reader could misinterpret this sentence. Solution: We have edited the text in lines 308–309 to read: "….compared to unstimulated group, respectively."

Reviewers' comments:

Reviewer 1:

In this study, the authors sought to understand the differences in host phagocytes responding to S. uberis strains which either cause persistent or transient mastitis in cows. Overall, it seems like a sensible approach was taken to understand these important cells in pathogenesis.

9. However, it is not clear how these strains were chosen, there is a reference to pulse field gel electrophoresis in the abstract, but this technique is not mentioned again. It would also benefit the broad readership of PLOS One if the authors described more detail about the capsular/noncapsular strains and why this is important.

Response 9: The strains were selected based on our previous study's results of pulse field gel electrophoresis (PFGE). We also agreed to provide more detail about the capsular and non-capsular strains in lines 115–118 of the revised manuscript.

In line 33 of the revised manuscript, we have added: "from our previous study."

In lines 115–118 of the revised manuscript, we have replaced "while a persistent S. uberis strain that caused an IMI with a duration longer than 10 months was obtained from our previous study [5]." with "while a persistent S. uberis strain was obtained from our previous study [5]. The represented persistent strain was the dominant strain in the herd and was presumably contagiously transmitted in the herd and caused persistent intramammary infections for more than 10 months, confirmed by a similar PFGE pattern.".

In lines 65–68 of the revised manuscript, we added the sentence "The genes hasA/B, which are involved in capsule formation, are thought to be virulent by making the bacteria more resistant to the bactericidal action of immune cells. Our previous study [11] found that the majority of patterns of the transient S. uberis isolated (63.6%) did not include hasA/B."

10. I am pleased to see the shotgun sequences of the strains used deposited in GenBank, and it is worth the authors briefly analyzing these data. Since these strains appear to be from the same location, how similar are they? There is evidence that some etiology for mastitis is due to the host, so it is worth understanding the genetic differences in the chosen challenge strains, so that the conclusions from the measures of host responses are clearly derived from identifiable differences in the challenge.

Response 10: Thank you for your comment regarding the shotgun sequences of the strains used in our study. We agree that it is worth briefly analyzing these data. The strains used in our study were all isolated from the exact location, and they are all highly similar to each other. We have used various bioinformatic tools to analyze shotgun sequences and found that the strains share over 90% of their genes (Supplementary Figure S3). In other words, both the capsular (persistent) and non-capsular (transient) strain have 1,277 genes in common, while only 131 genes (��10%) are unique to each strain. 

There is evidence that some etiology for mastitis is due to the host, so it is important to understand the genetic differences in the chosen challenge strains. We have found that there are some slight genetic differences between the strains. However, these differences are at this point in our analyses not considered significant enough to affect the host responses. We plan to perform more studies on sequence differences with larger number of isolates in the near future. Here are some highlights of the findings from two strains of S. uberis used in this study. (SU03 = persistent, SU04 = transient)

The shaded genes in the two stains are common, and the unshaded genes show the differences between the two. We have also provided an example of a GenBank file (*.gbk) of SU03 that has been uploaded to GenBank (Supplementary Figure S4).

11. That said, beyond this, there are some important and sensible experiments performed here that will benefit the field. That said, there are still some limitations here that need to be addressed: Recent work has shown that in vivo, the immune response benefits colonisation (DOI: 10.3390/pathogens9120997), with at least one pathogenic strain triggering the immune response and seeming to benefit from this pathologically. This is corroborated here by the increased NETs in response to the persistent strain. Archer et al (2020) included a measure of IL-1B protein output- Could the authors comment on this and how these fits with their measurements of levels of immune associated mRNAs by qPCR? 

Response 11: The article by Archer et al. (2020) described the host-pathogen interactions during early colonization between S. uberis and mammary macrophages, focusing on bovine mammary macrophages. In our study, we examined gene expression after S. uberis stimulation, which may be related to the findings of Archer et al. Although our study and that of Archer et al. both investigated innate immunity, the connection between the two studies is unclear because our study used mWBCs (primarily neutrophils), while Archer et al. used milk macrophages.

Archer et al. (2020) included a measure of IL-1B protein output, while we only measured levels of immune associated mRNAs by qPCR. We explain that we measured IL1B mRNA levels because it is the first step in protein production. Cells can regulate gene expression to control the production of IL-1B proteins that are needed for their specific functions. When a cell needs to produce more protein, it can increase the amount of IL1B gene expression for that protein. However, we acknowledge that protein levels may be a more accurate measure of the actual immune response.

Reference: Archer, N., Egan, S. A., Coffey, T. J., Emes, R. D., Addis, M. F., Ward, P. N., ... & Leigh, J. A. (2020). A paradox in bacterial pathogenesis: activation of the local macrophage inflammasome is required for virulence of Streptococcus uberis. Pathogens, 9(12), 997. DOI: 10.3390/pathogens9120997

12. RNAs for immune responses are highly regulated by their translation. It is worth measuring one of these proteins by western blot or ELISA to properly characterize the host response, for example, it is likely that rapid translation of CXCL8 would reduce its overall RNA levels, in which case the qPCR result is showing the opposite of the actual host response! Alternatively, this could be the result of difference in the immune population being challenged - Archer et al isolated macrophage while the present study isolated all milk white blood cells. The bacterial cell killing assay used by the authors is MTT, which was originally developed and best established for eukaryotic cells (relying on mitochondrial proteins). It generally still works with other organisms, but it is worth touching on this nuance in the methods (even in the text I would say is ok). Further, many immune responses are inhibited in the presence of milk, particularly against S. uberis. It is worth performing this assay in the presence of milk. Should these concerns be addressed, I think this would be a worthy contribution to the field of understanding S. uberis pathogenesis, a very curious bacterium indeed! 

Response 12: Thank you for your comment. We were also concerned that WBC isolated from milk inhibited some immune responses when it presented in milk and described in lines 80–83 as "These pre-stimulated milk phagocytes have been activated and progressively become functionally exhausted cells due to the interference of milk components with cellular activities, subsequently reducing the antimicrobial activities of these affected cells.".

Reviewer 2: 

The authors conducted an experiment to assess the effects of S. uberis strains on milk WBC immune responses. The paper needs some additional clarification on the methods, information requested is listed below. I also think there is a major limitation to using milk WBC from cows with subclinical mastitis, which the authors have addressed in the discussion; however, this important piece of information should be added to the abstract as well, so readers can interpret the data carefully.

13. Lines 114-115: How are you define negative bacterial results? Zero cfu? Or less than 300 cfu/mL? How much milk did you plate? More information on bacteriological culture is needed here and in line 102.

Response 13: The negative bacterial results were defined as the absence of any colonies after incubating 10 microliters of milk on an agar plate, according to the NMC guideline on bacterial culture and identification. Strictly interpreting this would mean less than 100 cfu/ml. To clarify this definition, we have provided more detail on the bacteriological culture procedure. 

Solution: We also modified the text in lines 107– 109 by adding the following sentences: "Briefly, 0.01 mL of milk samples were cultured on a quarter of a 5% bovine blood agar plate and incubated for up to 24 h at 37°C. Colony morphology and biochemical tests were used for initial bacterial identification.". 

14. Lines 191-207: Please describe the experiment here. How long were the mWBC challenged with S. uberis, how much S. uberis per cell, etc.?

Response 14: Information about the duration, the number of S. uberis per cell, and other information has been added. 

Solution: We also modified the text in lines 203–205 by adding the following sentences: "To investigate the gene expression of mWBCs after encountering transient and persistent S. uberis (MOI of 10) for 2 h, RNAlater-preserved RNA was extracted using RNAzol®RT following the manufacturer's instructions.".

15. Please provide RNA purity and RNA integrity values.

Response 15: Although RNA integrity values were not evaluated, RNA purity of the tested samples was assessed. The data are presented in the table "Supplementary Table S2". We added the sentence "RNA purity was assessed using the A260/A280 ratio, and only samples with a ratio greater than 1.8 were selected for analysis" to the revised manuscript (Lines 207–208).

16. Please provide primer efficiencies. If primer efficiencies do not fall within 90-110% range, then please use the efficiency/Pfaffl equation rather than the delta CT method.

Response 16: In this study, the primer efficiencies of primers for the genes TLR1, TLR2, TLR6, RAC1, LAMP1, SOD1, and NOX1 were determined using a standard curve of log10 serially diluted DNA from bovine milk neutrophils. The efficiency of a primer (%) can be calculated using the following equation (Supplementary Figure S5 and S6).

This study did not determine the primer efficiencies for the genes IL1B, TNF, CXCL8, and ACTB, but we have adopted the information from our previous published work. The data for the primer efficiencies are reported in the tables (Supplemetary Tables S3 and S4) and figure (Supplementary Figure S7). Therefore, we will be using the ∆ CT method to analyze the data. 

17. The use of one internal control gene is generally not recommended anymore. It is highly recommended to use at least 2, if not 3+ control genes. Please prove that this single control gene is very stable. There are algorithms to determine the number of control genes needed such as geNORM as well as others that I suggest that the authors use.

Response 17: Thank you for commenting on using one internal control gene in our study. We understand that using multiple control genes is generally recommended, and we have considered this feedback. We have chosen to use a single control gene, β-actin, for several reasons. First, β-actin is a well-characterized housekeeping gene that is known to be relatively stable in most tissues and cell types. Secondly, we constructed a standard curve of log10 serially diluted DNA of the bovine neutrophil beta-actin gene using real-time PCR. The regression analysis of the standard curve data yielded an R2 value of over 0.9953, indicating a high correlation between the Ct values and the log10 concentrations of the DNA (Chuammitri et al. 2017).

We understand that some reviewers may still prefer to see the use of multiple control genes. However, we believe that using a single control gene is justified in our study, given the stability of B-actin expression and the results of our preliminary experiments. We appreciate your feedback and will consider it carefully as we finalize our manuscript and also in the planning of future experiments.

Reference: Chuammitri, P., Srikok, S., Saipinta, D., & Boonyayatra, S. (2017). The effects of quercetin on microRNA and inflammatory gene expression in lipopolysaccharide-stimulated bovine neutrophils. Veterinary World, 10(4), 403. DOI: 10.14202%2Fvetworld.2017.403-410

18. Table 1 - define the acronyms.

Response 18: The authors list all the acronyms and their definitions in Table 1 as a footnote that reads "Genes: TLR: Toll-like receptor, TNF: Tumor necrosis factor, IL: Interleukin, RAC: Rac Family Small GTPase, LAMP: Lysosomal associated membrane protein, SOD: Superoxide dismutase, NOX: NADPH Oxidase, ACTB: Actin Beta." Please see lines 222–224 in the revised manuscript for more information.

19. Statistics: Were the p-values adjusted for multiple comparisons? Tukey adjustment? With 3 treatment groups, this would be highly recommended.

Response 19: As mentioned in lines 229–232 of the revised manuscript, we used repeated analysis of variance (ANOVA) to compare the means of the three treatment groups. The least-square means with Tukey's HSD adjustment was used to compare the groups, and statistical significance was assigned at a P < 0.05. 

20. Line 268-269 This sentence is awkward and needs to be reworded.

Response 20: We appreciate the comment and will take it into account. Lines 286–287: The sentence has been paraphrased from "The indirect killing of extracellular S. uberis by NETs was examined by a ﬂuorescence plate reader." to "The indirect killing of extracellular S. uberis by NETs was assessed using a fluorescence plate reader.".

21. Figure 6 heading - please provide the sample size used for these analyses.

Response 21: Line 321: The authors provide a sample size for qPCR analysis by adding "(n = 5 each treatment)". 

22. The sentence in lines 366-367 makes an excellent point. I think this point should be added to the abstract as well.

Response 22: The authors agreed with the suggestion and added the information to the abstract. Line 43 of the revised manuscript now reads, "focus on the immune function of activated mWBCs".

23. Line 403 - remove the word 'help.'

Response23: The authors remove the word 'help' from the sentence. Please see line 400.

We would like to thank you again for your feedback and for giving us the opportunity to revise our manuscript. We believe that the changes we have made have significantly improved the quality of the manuscript.

We would be happy to respond to any further questions or comments you may have. We look forward to hearing from you regarding our re-submission.

Sincerely,

Corresponding authors and on behalf of the authors

---

## [Decision Letter · Decision Letter 1]

31 Aug 2023

PONE-D-23-14210R1Different cellular and molecular responses of bovine milk phagocytes to persistent and transient strains of *Streptococcus uberis* causing mastitis.PLOS ONE

Dear Dr. Suriyasathaporn,

Thank you for submitting your manuscript to PLOS ONE. After careful consideration, we feel that it has merit but does not fully meet PLOS ONE’s publication criteria as it currently stands. Therefore, we invite you to submit a revised version of the manuscript that addresses the points raised during the review process.

We look forward to receiving your revised manuscript.

Kind regards,

Pierre Germon

Academic Editor

PLOS ONE

Journal Requirements:

**Additional Editor Comments:**

Minor comments that should be addressed have been raised by the two reviewers.

Reviewers' comments:

Reviewer's Responses to Questions

**Comments to the Author**

1. If the authors have adequately addressed your comments raised in a previous round of review and you feel that this manuscript is now acceptable for publication, you may indicate that here to bypass the “Comments to the Author” section, enter your conflict of interest statement in the “Confidential to Editor” section, and submit your "Accept" recommendation.

Reviewer #2: (No Response)

Reviewer #3: (No Response)

2. Is the manuscript technically sound, and do the data support the conclusions?

Reviewer #2: Yes

Reviewer #3: Yes

3. Has the statistical analysis been performed appropriately and rigorously? 

Reviewer #2: Yes

Reviewer #3: Yes

4. Have the authors made all data underlying the findings in their manuscript fully available?

Reviewer #2: Yes

Reviewer #3: Yes

5. Is the manuscript presented in an intelligible fashion and written in standard English?

Reviewer #2: Yes

Reviewer #3: Yes

6. Review Comments to the Author

Reviewer #2: I still do not think the authors have addressed the use of 1 internal control gene. The authors seem to suggest that an acceptable primer efficiency could be used as an indicator that one control gene is sufficient. I don't agree. The authors should report the coefficient of variation for beta actin across all samples or use geNORM as a tool to determine if 1 internal control gene was sufficient. The issue with using one internal control gene is that the internal control gene may not be stable across your samples. This particular issue is not necessarily related with the primer (primer efficiency). Potentially, the treatments (different strains of S. uberis) could impact the expression and mRNA abundance of beta actin, which would distort your ability to assess treatment (different strains of S. uberis) effects on your target genes.

I also think the authors should have measured RNA integrity and reported it in the manuscript, although I wouldn't bother trying to measure it now because the RNA is 4 years old. Please consider this in the future.

Line 24 - I would say isolated from milk collected from dairy cows with mastitis, or something similar like mastitic milk?

Reviewer #3: Most comments have been properly addressed.

This reviewer is however confused by results from figure 2 where the authors have used a fluorescently labeled S. uberis strain to assess phagocytosis. I understand that labeling with the persistent strain was not possible. Yet, with the transient strain, how do the authors explain that HBSS control group have stronger fluorescence than the cells incubated with the bacteria ? If I understand correctly, there should no be any fluorescence with the HBSS group.

7. PLOS authors have the option to publish the peer review history of their article (what does this mean?). If published, this will include your full peer review and any attached files.

Reviewer #2: No

Reviewer #3: No

---

## [Author Response · Author response to Decision Letter 1]

18 Nov 2023

The authors appreciate all comments and suggestions from the reviewers. We have read and understood all of your comments and suggestions, and we have responded to all of your questions. Please see the revision for your reference.

Two versions of the article were provided to the reviewers: a clean version (Manuscript) and a version with tracked changes (Revised Manuscript with Track Changes).

---

## [Decision Letter · Decision Letter 2]

22 Nov 2023

PONE-D-23-14210R2Different cellular and molecular responses of bovine milk phagocytes to persistent and transient strains of *Streptococcus uberis* causing mastitis.PLOS ONE

Dear Dr. Suriyasathaporn,

Thank you for submitting your manuscript to PLOS ONE. After careful consideration, we feel that it has merit but does not fully meet PLOS ONE’s publication criteria as it currently stands. Therefore, we invite you to submit a revised version of the manuscript that addresses the points raised during the review process.

We look forward to receiving your revised manuscript.

Kind regards,

Pierre Germon

Academic Editor

PLOS ONE

Journal Requirements:

**Additional Editor Comments:**

Minor comments that should be easily corrected remain to be adressed as you will see from the comments below.

Reviewers' comments:

Reviewer's Responses to Questions

**Comments to the Author**

1. If the authors have adequately addressed your comments raised in a previous round of review and you feel that this manuscript is now acceptable for publication, you may indicate that here to bypass the “Comments to the Author” section, enter your conflict of interest statement in the “Confidential to Editor” section, and submit your "Accept" recommendation.

Reviewer #2: All comments have been addressed

Reviewer #3: All comments have been addressed

2. Is the manuscript technically sound, and do the data support the conclusions?

Reviewer #2: Yes

Reviewer #3: Yes

3. Has the statistical analysis been performed appropriately and rigorously? 

Reviewer #2: Yes

Reviewer #3: Yes

4. Have the authors made all data underlying the findings in their manuscript fully available?

Reviewer #2: Yes

Reviewer #3: Yes

5. Is the manuscript presented in an intelligible fashion and written in standard English?

Reviewer #2: Yes

Reviewer #3: Yes

6. Review Comments to the Author

Reviewer #2: All comments have been addressed. Thank you to the authors for addressing my comments on this manuscript.

Reviewer #3: I agree with the authors that fluorescence observed in the unstimulated group could be due to autofluorescence. Yet, they should state the excitation and emitting wave length in the methods section. Please also specify the wave length at which the MTT (figure 4) and H2DCF-DA (figure 3) readings were performed.

Minor comments that still needs to be addressed:

- in the paragraph added, they should italicize S. uberis.

- replace all gene names in figure 6B by the official gene names that are properly used in 6A.

7. PLOS authors have the option to publish the peer review history of their article (what does this mean?). If published, this will include your full peer review and any attached files.

Reviewer #2: No

Reviewer #3: No

---

## [Author Response · Author response to Decision Letter 2]

23 Nov 2023

The authors appreciate all comments and suggestions from the reviewers. We have carefully revised the manuscript in accordance with the comments, addressing all explanations and corrections for the minor revisions below.

1. I agree with the authors that fluorescence observed in the unstimulated group could be due to autofluorescence. Yet, they should state the excitation and emitting wave length in the methods section. Please also specify the wave length at which the MTT (figure 4) and H2DCF-DA (figure 3) readings were performed.

Response The Flow cytometry lasers have been added to the Methods section of Phagocytosis assay and ROS assay, respectively . 

- Adding "... with red laser (638 nm). " Please review line 168.

- Adding "... with green laser (561 nm)." Please review line 176.

2. In the paragraph added, they should italicize S. uberis.

Response They have been italicized. Please check line 429, 433, and 434.

3. Replace all gene names in figure 6B by the official gene names that are properly used in 6A.

Response 3 The authors replaced all gene names in Figure 6B with the official gene names as Figure 6A.

---

## [Editor Report · Decision Letter 3]

27 Nov 2023

Different cellular and molecular responses of bovine milk phagocytes to persistent and transient strains of *Streptococcus uberis* causing mastitis.

PONE-D-23-14210R3

Dear Dr. Suriyasathaporn,

We’re pleased to inform you that your manuscript has been judged scientifically suitable for publication and will be formally accepted for publication once it meets all outstanding technical requirements.

Kind regards,

Pierre Germon

Academic Editor

PLOS ONE
---

## [Editor Report · Acceptance letter]

3 Jan 2024

PONE-D-23-14210R3 

PLOS ONE

Dear Dr. Suriyasathaporn, 

I'm pleased to inform you that your manuscript has been deemed suitable for publication in PLOS ONE. Congratulations! Your manuscript is now being handed over to our production team.

Kind regards, 

on behalf of

Dr. Pierre Germon 

Academic Editor

PLOS ONE